# Computational Separations between Sampling and Optimization

Kunal Talwar

Google Brain
Mountain View, CA
kunal@google.com

## Abstract

Two commonly arising computational tasks in Bayesian learning are Optimization (Maximum A Posteriori estimation) and Sampling (from the posterior distribution). In the convex case these two problems are efficiently reducible to each other. Recent work [Ma et al., 2019] shows that in the non-convex case, sampling can sometimes be provably faster. We present a simpler and stronger separation. We then compare sampling and optimization in more detail and show that they are provably incomparable: there are families of continuous functions for which optimization is easy but sampling is NP-hard, and vice versa. Further, we show function families that exhibit a sharp phase transition in the computational complexity of sampling, as one varies the natural temperature parameter. Our results draw on a connection to analogous separations in the discrete setting which are well-studied.

## 1 Introduction

Given a a compact set $\mathcal{X} \subseteq \mathbb{R}^d$ and function $f : \mathcal{X} \to \mathbb{R}$, one can define two natural problems:

**Optimize**$(f, \mathcal{X}, \varepsilon)$ : Find $\mathbf{x} \in \mathcal{X}$ such that $f(\mathbf{x}) \leq f(\mathbf{x}') + \varepsilon$ for all $\mathbf{x}' \in \mathcal{X}$.

**Sample**$(f, \mathcal{X}, \eta)$ : Sample from a distribution on $\mathcal{X}$ that is $\eta$-close to $\mu^\star(\mathbf{x}) \propto \exp(-f(\mathbf{x}))$.

These problems arise naturally in machine learning settings. When $f$ is the negative log likelihood function of a posterior distribution, the optimization problem corresponds to the Maximum A Posteriori (MAP) estimate, whereas the Sampling problem gives us a sample from the posterior. In this work we are interested in the computational complexities of these tasks for specific families of functions.

When $f$ and $\mathcal{X}$ are both convex, these two problems have a deep connection (see e.g. Lovasz and Vempala [2006]) and are efficiently reducible to each other in a very general setting. There has been considerable interest in both these problems in the non-convex setting. Given that in practice, we are often able to practically optimize certain non-convex loss functions, it would be appealing to extend this equivalence beyond the convex case. If sampling could be reduced to optimization for our function of interest (e.g. differentiable Lipschitz functions), that might allow us to design sampling algorithms for the function that are usually efficient in practice. Ma et al. [2019] recently showed that in the case when $f$ is not necessarily convex (and $\mathcal{X} = \mathbb{R}^d$), these problems are not equivalent. They exhibit a family of continuous, differentiable functions for which approximate sampling can be done efficiently, but where approximate optimization requires exponential time (in an oracle model *à la* Nemirovsky and Yudin [1983]). In this work, we study the relationship of these two problems in more detail.

To aid the discussion, it will be convenient to consider a more general sampling problem where we want to sample with probability proportional to $\exp(-\lambda f(\mathbf{x}))$ for a parameter $\lambda > 0$. Such a scaling has no effect on the optimization problem, up to scaling of $\varepsilon$. However changing $\lambda$ can signifcantly change the distribution for the sampling problem. In statistical physics literature, this parameter is the inverse temperature. For families $\mathcal{F}$ that are invariant to multiplication by a positive scalar (such as the family of convex functions), this $\lambda$ parameter has no impact on the complexity of sampling from the family. We will however be looking at families of functions that are controlled in some way (e.g. bounded, Lipschitz, or Smooth) and do not enjoy such an invariance to scale. E.g. in some Bayesian settings, each sample may give us a 1-smooth negative log likelihood function, so we may want to consider the family $\mathcal{F}_{smooth}$ of 1-smooth functions. Given $n$ i.i.d. samples, the posterior log likelihood would be $-n\overline{f}$, where $\overline{f} = \frac{1}{n}\sum_i f_i$ is in $\mathcal{F}_{smooth}$. The parameter $\lambda$ then corresponds naturally to the number of samples $n$.

This phenomenon of sampling being easier than optimization is primarily a "high temperature" or "low signal" phenomenon. As $\lambda$ approaches infinity, the distribution $\exp(-\lambda f)$ approaches a point mass at the minimizer of $f$. This connection goes back to at least Kirkpatrick et al. [1983] and one can easily derive a quantitative finite-$\lambda$ version of this statement for many function families. Ma et al. [2019] reconcile this with their separation by pointing out that their sampling algorithm becomes inefficient as $\lambda$ increases.

We first show a more elementary and stronger separation. We give a simple family of continuous Lipschitz functions which are efficiently samplable but hard even to approximately optimize. This improves on the separation in Ma et al. [2019] since our sampler is exact (modulo representation issues), and much simpler. The hardness of optimization here is in the oracle model, where the complexity is measured in terms of number of point evaluations of the function or its gradients.

While these oracle model separations rule out black-box optimization, they leave open the possibility of efficient algorithms that access the function in a different way. We next show that this hardness can be strengthened to an NP-hardness for an efficiently computable $f$. This allows for the implementation of any oracle for $f$ or its derivatives. Thus assuming the Exponential Time Hypothesis [Impagliazzo and Paturi, 2001], our result implies the oracle model lower bounds. Additionally, it rules out efficient non-blackbox algorithms that could examine the implementation of $f$ beyond point evaluations. We leave open the question of whether other oracle lower bounds [Nemirovsky and Yudin, 1983, Nesterov, 2014, Bubeck, 2015, Hazan, 2016] in optimization can be strengthened to NP-hardness results.

We next look at the large $\lambda$ case. As discussed above, for large enough $\lambda$ sampling must be hard if optimization is. *Is hardness of optimization the only obstruction to efficient sampling?* We answer this question in the negative. We exhibit a family of functions for which optimization is easy, but where sampling is NP-hard for large enough $\lambda$. We draw on extensive work on the discrete analogs of these questions, where $f$ is simple (e.g. linear) but $\mathcal{X}$ is a discrete set.

Our upper bound on optimization for this family can be strengthened to work under weaker models of access to the function, where we only have blackbox access to the function. In other words, there are functions that can be optimized via gradient descent for which sampling is NP-hard. Conceptually, this separation is a result of the fact that finding one minimizer suffices for optimization whereas sampling intuitively requires finding *all* minima.

Both the separation result of Ma et al. [2019], and our small-$\lambda$ result have the property that the sampling algorithm's complexity increases exponentially in $poly(\lambda)$. Thus as we increase $\lambda$, the problem gradually becomes harder. *Is there always a smooth transition in the complexity of sampling?*

Our final result gives a surprising negative answer. We exhibit a family of easy-to-optimize functions for which there is a sharp threshold: there is a $\lambda_c$ such that for $\lambda < \lambda_c$, sampling can be done efficiently, whereas for $\lambda > \lambda_c$, sampling becomes NP-hard. In the process, this demonstrates that for some families of functions, efficient sampling algorithms can be very structure-dependent, and do not fall into the usual Langevin-dynamics, or rejection-sampling categories.

Our results show that once we go beyond the convex setting, the problems of sampling and optimization exhibit a rich set of computational behaviors. The connection to the discrete case should help further understand the complexities of these problems.

The rest of the paper is organized as follows. We start with some preliminaries in Section 2. We give a simple separation between optimization and sampling in Section 3 and derive a computational version of this separation. Section 4 relates the discrete sampling/optimization problems on the hypercube to their continuous counterparts, and uses this connection to derive NP-hardness results for sampling for function families where optimization is easy. In Section 5, we prove the sharp threshold for $\lambda$. We describe additional related work in Section 6. Some missing proofs and strengthenings of our results are deferred to the supplementary material.

## 2  Preliminaries

We consider real-valued functions $f : \mathbb{R}^d \to \mathbb{R}$. We will be restricting ourselves to functions that are continuous and bounded. We say a function $f$ is $L$-Lipschitz if $f(\mathbf{x}) - f(\mathbf{x}') \leq L \cdot \|\mathbf{x} - \mathbf{x}'\|$ for all $\mathbf{x}, \mathbf{x}' \in \mathbb{R}^d$. In this work, $\|\cdot\|$ will denote the Euclidean norm.

We will look at specific families of functions which have compact representations, and ask questions about efficiency of optimization and sampling. We will think of $d$ as a parameter, and look at function families such that at any function in the family can be computed in $poly(d)$ time and space.

We will look at constrained optimization in this work and our constraint set $\mathcal{X}$ will be a Euclidean ball. Our hardness results however do not stem from the constraint set, and nearly all of our results can be extended easily to the unconstrained case.

Given $\lambda > 0$ and a function $f$, we let $\mathcal{D}_f^{\lambda, \mathcal{X}}$ denote the distribution on $\mathcal{X}$ with $\Pr[\mathbf{x}] \propto \exp(-\lambda f(\mathbf{x}))$. When $\mathcal{X}$ is obvious from context, we will usually omit it and write $\mathcal{D}_f^{\lambda}$. We will $Z_f^{\lambda, \mathcal{X}}$ for the *partition function* $\int_{\mathcal{X}} \exp(-\lambda f(\mathbf{x})) \mathrm{d}\mathbf{x}$.

We will also look at real-valued functions $h : \mathbb{H}_d \to \mathbb{R}$, where $\mathbb{H}_d = \{-1, 1\}^d$ is the $d$-dimensional hypercube. We will often think of a $\mathbf{y} \in \mathbb{H}_d$ as being contained in $\mathbb{R}^d$. Analogous to the Euclidean space case, we define $\mathcal{D}_h^{\lambda, \mathbb{H}_d}$ as the distribution over the hypercube with $\Pr[\mathbf{y}] \propto \exp(-\lambda h(\mathbf{y}))$, and define $Z_h^{\lambda, \mathbb{H}_d}$ to be $\sum_{\mathbf{y} \in \mathbb{H}_d} \exp(-\lambda h(\mathbf{y}))$.

We say that an algorithm $\eta$-samples from $\mathcal{D}_f^{\lambda}$ if it samples from a distribution that is $\eta$-close to $\mathcal{D}_f^{\lambda}$ in statistical distance. We will also use the *Wasserstein* distance between distributions on $\mathbb{R}^d$, defined as $\mathcal{W}(P, Q) \overset{eq}{=} \inf_\pi \mathbb{E}_{(\mathbf{x}, \mathbf{x}') \sim \pi}[\|\mathbf{x} - \mathbf{x}'\|_2]$, where the $\inf$ is taken over all couplings $\pi$ between $P$ and $Q$. We remark that our results are not sensitive to the choice of distance between distributions and extend in a straightforward way to other distances. As is common in literature on sampling from continuous distributions, we will for the most part assume that we can sample from a real-valued distribution such as a Gaussian and ignore bit-precision issues. Statements such as our NP-hardness results usually require finite precision arithmetic. This issue is discussed at length by Tosh and Dasgupta [2019] and following them, we will discuss using Wasserstein distance in those settings. An $\varepsilon$-optimizer of $f$ is a point $\mathbf{x} \in \mathcal{X}$ such that $f(\mathbf{x}) \leq f(\mathbf{x}') + \varepsilon$ for any $\mathbf{x}' \in \mathcal{X}$.

In the supplementary material, we quantify the folklore results showing that sampling for high $\lambda$ implies approximate optimization. Quantitatively, they say that for $L$-Lipschitz functions over a ball of radius $R$, sampling implies $\varepsilon$-approximate optimization if $\lambda \geq \Omega(d \ln \frac{LR}{\varepsilon} / \varepsilon)$. Similarly, for $\beta$-smooth functions over a ball of radius $R$, $\lambda \geq \Omega(d \ln \frac{\beta R}{\varepsilon} / \varepsilon)$ suffices to get $\varepsilon$-close to an optimum.

## 3  A Simple separation

We consider the case when $\mathcal{X} = B_d(\mathbf{0}, 1)$ is the unit norm Euclidean ball in $d$ dimensions. We let $\mathcal{F}_{Lip}$ be the family of all 1-lipschitz functions from $\mathcal{X}$ to $\mathbb{R}$. We show that for any $f \in \mathcal{F}_{Lip}$, exact sampling can be done in time $\exp(O(\lambda))$. On the other hand, for any algorithm, there is an $f \in \mathcal{F}_{Lip}$ that forces the algorithm to use $\Omega(\lambda/\varepsilon)^d$ queries to $f$ to get an $\varepsilon$-approximate optimal solution. Thus, e.g., for constant $\lambda$, sampling can be done in $poly(d)$ time, whereas optimization requires time exponential in the dimension. Moreover, for any $\lambda \leq d$, there is an exponential gap between the complexities of these two problems. Our lower bound proof is similar to the analagous claim in Ma et al. [2019], but has better parameters due to the simpler setting. Our upper bound proof is signifantly simpler and gives an exact sampler.

**Theorem 1** (Fast Sampling). *There is an algorithm that for any $f \in \mathcal{F}_{Lip}$, outputs a sample from $\mathcal{D}_f^\lambda$ and makes an expected $O(\exp(2\lambda))$ oracle calls to computing $f$.*

*Proof.* The algorithm is based on rejection sampling. We first compute $f(\mathbf{0})$ and let $M = f(\mathbf{0}) - 1$. By the Lipschitzness assumption, $f(\mathbf{x}) \in [M, M+2]$ for all $\mathbf{x}$ in the unit ball. The algorithm now repeatedly samples a random point $\mathbf{x}$ from the unit ball. With probability $\exp(\lambda(M - f(\mathbf{x})))$, this point is accepted and we output it. Otherwise we continue.

Since $\exp(\lambda(M - f(\mathbf{x}))) \in [\exp(-2\lambda), 1]$, this is a rejection sampling algorithm, and it outputs an exact sample from $\mathcal{D}_f^\lambda$. Each step accepts with probability at least $\exp(-\lambda)$. Thus the algorithm terminates in an expected $O(\exp(2\lambda))$ many steps, each of which requires one evaluation of $f$. $\square$

*Remark* 1. The above algorithm assumes access to an oracle to sample from $B_d(\mathbf{0}, 1)$ to arbitrary precision. This can be replaced by sampling from a grid of finite precision points in the ball. This creates two sources of error. Firstly, the function is not constant in the grid cell. This error is easily bounded since $f$ is Lipschitz. Secondly, some grid cells may cross the boundary of $B_d(\mathbf{0}, 1)$. This is a probability $d2^{-b}$ event when sampling a grid point with $b$ bits of precision. Taking these errors into account gives us a sample within Wasserstein distance at most $O((d + \lambda)2^{-b})$.

*Remark* 2. The above is a Las Vegas algorithm. One can similarly derive a Monte Carlo algorithm by aborting and outputting a random $\mathbf{x}$ after $\exp(2\lambda) \log \frac{1}{\eta}$ steps.

*Remark* 3. Under the assumptions in Ma et al. [2019] ($\beta$-smooth $f$, $\nabla(\mathbf{0}) = \mathbf{0}$, $\kappa\beta$-strong convexity outside a ball of radius $R$), a direct reduction to our setting will be lossy and a rejection-sampling-based approach will not be efficient. The Langevin dynamics based sampler in that work is more efficient under their assumptions.

**Theorem 2** (No Fast Optimization). *For any algorithm $\mathcal{A}$ that queries $f$ or any of its derivatives at less than $(1/4\varepsilon)^d$ points, there is an $f \in \mathcal{F}_{Lip}$ for which $\mathcal{A}$ fails to output an $\varepsilon$-optimizer of $f$ except with negligible probability.*

*Proof.* Consider the function $f_\mathbf{x}$ that is zero everywhere, except for a small ball of radius $2\varepsilon$ around $\mathbf{x}$, where it is $f(\mathbf{y}) = \|\mathbf{y} - \mathbf{x}\| - 2\varepsilon$. i.e. the function[1] is $f_\mathbf{x}(\mathbf{y}) = \min(0, \|\mathbf{y} - \mathbf{x}\| - 2\varepsilon)$. This function has optimum $-2\varepsilon$. Let $g$ be the zero function.

Let $\mathcal{A}$ be an algorithm (possibly randomized) that queries $f$ or its derivatives at $T \le (1/4\varepsilon)^d$ points. Consider running $\mathcal{A}$ on a function $f$ chosen randomly as:

$$f = \begin{cases} g & \text{with probability } \frac{1}{2} \\ f_\mathbf{x} & \text{for a } \mathbf{x} \text{ chosen u.a.r. from } B(\mathbf{0}, 1) \text{ otherwise.} \end{cases}$$

Until $\mathcal{A}$ has queried a point in $B(\mathbf{x}, 2\varepsilon)$, the behavior of the algorithm on $f_\mathbf{x}$ and $g$ is identical, since the functions and all their derivatives agree outside this ball. Since an $\varepsilon$-approximation must distinguish these two cases, for $\mathcal{A}$ to succeed, it must query this ball. The probability that $\mathcal{A}$ queries in this ball in any given step is at most $\frac{\text{vol}(B(\mathbf{x}, 2\varepsilon))}{\text{vol}(B(\mathbf{0}, 1))} = (2\varepsilon)^d$. As $\mathcal{A}$ makes only $(1/4\varepsilon)^d$ queries in total, the expected number of queries $\mathcal{A}$ makes to the ball $B(\mathbf{x}, 2\varepsilon)$ is at most $2^{-d}$. Thus with probability at least $1 - \frac{1}{2^d}$, the algorithm fails to distinguish $g$ from $f_\mathbf{x}$, and hence cannot succeed. $\square$

## 3.1 Making the Separation Computational

The oracle-hardness of Theorem 2 stems from possibly "hiding" a minimizer $\mathbf{x}$ of $f$. The computational version of this hardness result will instead possibly hide an appropriate encoding of a satisfying assignment to a 3SAT formula.

**Theorem 3** (No Fast Optimization: Computational). *There is a constant $\varepsilon > 0$ such that it is NP-hard to $\varepsilon$-optimize an efficiently computable Lipschitz function over the unit ball.*

*Proof.* Let $\phi : \{0,1\}^d \to B_d(\mathbf{0}, 1)$ be a map such that $\phi$ is efficiently computable, $\|\phi(\mathbf{y}) - \phi(\mathbf{y}')\| \ge 4\varepsilon$ and such that given $\mathbf{x} \in B_d(\phi(\mathbf{y}), 2\varepsilon)$, we can efficiently recover $\mathbf{y}$. For a small enough absolute

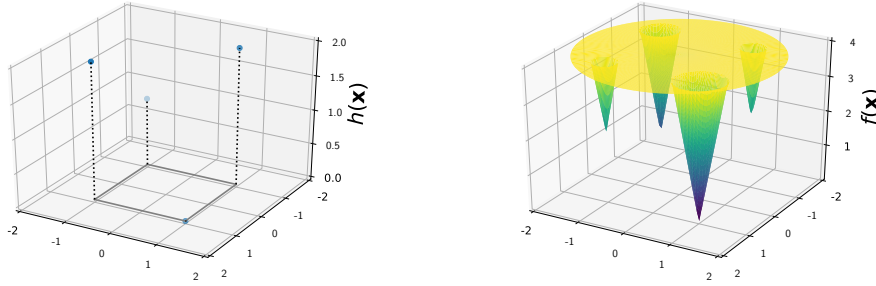

Figure 1: (Left) An example of a function $h$ for $d = 2$, with the 2-d hypercube shown in gray and the values of $h$ denoted by blue points. (Right) The corresponding function $f$ that results from the transformation, for $M = 4, R = 2$.

constant $\varepsilon$, such maps can be easily constructed using error correcting codes and we defer details to supplementary material.

We start with an an instance $I$ of 3SAT on $d$ variables, and define $f$ as follows. Given a point $\mathbf{x}$, we first find a $\mathbf{y} \in \{0, 1\}^d$, if any, such that $\mathbf{x} \in B_d(\phi(\mathbf{y}), 2\varepsilon)$. If no such $\mathbf{y}$ exists, $f(\mathbf{x})$ is set to 0. If such a $\mathbf{y}$ exists, we interpret it as an assignment to the variables in the 3SAT instance $I$ and set $f(\mathbf{x})$ to be $\min(0, \|\phi(\mathbf{y}) - \mathbf{x}\| - 2\varepsilon)$ if $\mathbf{y}$ is a satisfying assignment to instance $I$, and to 0 otherwise.

It is clear from definition that $f$ as defined is efficiently computable. Moreover, the minimum attained value for $f$ is $-2\varepsilon$ if $I$ is satsifiable, and 0 otherwise. Since distinguishing between these two cases is NP-hard, so is $\varepsilon$-optimization of $f$. $\qquad\square$

We note that assuming the exponential time hypothesis, this implies the $\exp(\Omega(d))$ oracle complexity lower bound of Theorem 2.

## 4 Relating Discrete and Continuous Settings

For any function $h$ on the hypercube, we can construct a function on $f$ on $\mathbb{R}^d$ such that optimization of $f$ and $h$ are reducible to each other, and similarly sampling from $f$ and $h$ are reducible to each other. This would allow us to use separation results for the hypercube case to establish analagous separation results for the continuous case.

**Theorem 4.** *Let $h : \mathbb{H}_d \to \mathbb{R}$ have range $[0, d]$. Fix $M \geq 2d, R \geq 2\sqrt{d}$. Then there is a funtion $f : B_d(0, R) \to \mathbb{R}$ satisfying the following properties:*

**Efficiency:** *Given $\mathbf{x} \in B_d(0, R)$ and oracle access to $h$, $f$ can computed in polynomial time.*

**Lipschitzness:** *$f$ is continuous and $L$-Lipschitz for $L = 2M$.*

**Sampler Equivalence:** *Fix $\lambda \geq \frac{4d \ln 24R}{M}$. Given access to an $\eta$-sampler for $\mathcal{D}_h^{\lambda, \mathbb{H}_d}$, there is an efficient $\eta'$-sampler for $\mathcal{D}_f^{\lambda}$, for $\eta' = \eta + \exp(-\Omega(d))$. Conversely, given access to an $\eta$-sampler for $\mathcal{D}_f^{\lambda}$, there is an efficient $\eta'$-sampler for $\mathcal{D}_h^{\lambda, \mathbb{H}_d}$ for $\eta' = \eta + \exp(-\Omega(d))$.*

*Proof.* The function $f$ is fairly natural: it takes a large value $M \geq 2d$ at most points, except in small balls around the hypercube vertices. At each hypercube vertex, $f$ is equal to the $h$ value at the vertex, and we interpolate linearly in a small ball. See Figure 2 for an illustration.

Formally, let $\text{round} : \mathbb{R} \to \{-1, 1\}$ be the function that takes the value 1 for $x \geq 0$ and $-1$ otherwise, and let $\text{round} : \mathbb{R}^d \to \mathbb{H}_d$ be its natural vectorized form that applies the function coordinate-wise. Let $g(\mathbf{x}) = \|\mathbf{x} - \text{round}(\mathbf{x})\|$ denote the Euclidean distance from $\mathbf{x}$ to $\text{round}(\mathbf{x})$. Let $s = 2M$. The

function $f$ is defined as follows:

$$f(\mathbf{x}) = \begin{cases} h(\text{round}(\mathbf{x})) + s \cdot g(\mathbf{x}) & \text{if } g(\mathbf{x}) \leq \frac{M - h(\text{round}(\mathbf{x}))}{s} \\ M & \text{if } g(\mathbf{x}) \geq \frac{M - h(\text{round}(\mathbf{x}))}{s} \end{cases}$$

It is easy to verify that $f$ is continuous. Moreover, since $M \geq 2d$, and $h$ has range $[0, d]$, the value $\frac{M - h(\text{round}(\mathbf{x}))}{s}$ is in the range $[\frac{1}{4}, \frac{1}{2}]$. It follows that $f$ takes the value $M$ outside balls of radius $\frac{1}{2}$ around the hypercube vertices, and is strictly smaller than $M$ in balls for radius $\frac{1}{4}$.

Since $\text{round}(\mathbf{x})$ is easy to compute, this implies that $f$ can be computed in polynomial time, using a single oracle call to $h$. Moreover it is immediate from the definition that the function $f$ has Lipschitz constant $s$.

Note that $f(\mathbf{y}) = h(\mathbf{y})$ for $\mathbf{y} \in \mathbb{H}_d$ and $f(\mathbf{x}) \geq h(\text{round}(\mathbf{x}))$ for all $\mathbf{x} \in B_d(0, R)$. This implies that the minimum value of $f$ is the same as the minimum value of $h$, and indeed any ($\varepsilon$-)minimizer $\mathbf{y}$ of $h$ also ($\varepsilon$-)minimizes $f$. Conversely, let $\mathbf{x}$ be an $\varepsilon$-minimizer of $f$. Since $h(\text{round}(\mathbf{x})) \leq f(\mathbf{x})$, it follows that $\text{round}(\mathbf{x})$ is an $\varepsilon$-minimizer of $h$.

Finally we prove the equivalence of approximate sampling. Towards that goal, we define an intermediate distribution on $B_d(0, R)$. Let $\widehat{\mathcal{D}_f^\lambda}$ be the distribution $\mathcal{D}_f^\lambda$ conditioned on being in $\cup_{\mathbf{y} \in \mathbb{H}_d} B_d(\mathbf{y}, \frac{1}{4})$.

We first argue that $\eta$-samplability of $\widehat{\mathcal{D}_f^\lambda}$ is equivalent to $\eta$-samplability of $\mathcal{D}_h^{\lambda, \mathbb{H}_d}$. Suppose that $X$ is a sample from $\widehat{\mathcal{D}_f^\lambda}$. Then for any $\mathbf{y}^\star \in \mathbb{H}_d$,

$$\begin{aligned} \Pr[X \in B_d(\mathbf{y}^\star, \frac{1}{4})] &= \frac{\int_{B_d(\mathbf{y}^\star, \frac{1}{4})} \exp(-\lambda f(\mathbf{x})) \, d\mathbf{x}}{\sum_{\mathbf{y} \in \mathbb{H}_d} \int_{B_d(\mathbf{y}, \frac{1}{4})} \exp(-\lambda f(\mathbf{x})) \, d\mathbf{x}} \\ &= \frac{\exp(-\lambda h(\mathbf{y}^\star)) \cdot \int_{B_d(\mathbf{y}^\star, \frac{1}{4})} \exp(-s\lambda g(\mathbf{x})) \, d\mathbf{x}}{\sum_{\mathbf{y} \in \mathbb{H}_d} \exp(-\lambda h(\mathbf{y})) \cdot \int_{B_d(\mathbf{y}, \frac{1}{4})} \exp(-s\lambda g(\mathbf{x})) \, d\mathbf{x}} \\ &= \frac{\exp(-\lambda h(\mathbf{y}^\star))}{\sum_{\mathbf{y} \in \mathbb{H}_d} \exp(-\lambda h(\mathbf{y}))} \end{aligned}$$

Thus $\text{round}(X)$ is a sample from $\mathcal{D}_h^{\lambda, \mathbb{H}_d}$. Conversely, the same calculation implies that given a sample $Y$ from $\mathcal{D}_h^{\lambda, \mathbb{H}_d}$, and a vector $Z \in B_d(\mathbf{0}, \frac{1}{4})$ sampled proportional to $\exp(-s\lambda \|\mathbf{z}\|)$, $Y + Z$ is a sample from $\widehat{\mathcal{D}_f^\lambda}$. Noting that $Z$ is a sample from a efficiently sample-able log-concave distribution completes the equivalence.

We next argue that $\mathcal{D}_f^\lambda$ and $\widehat{\mathcal{D}_f^\lambda}$ are $\exp(-\Omega(d))$-close as distributions. Since $\widehat{\mathcal{D}_f^\lambda}$ is a conditioning of $\mathcal{D}_f^\lambda$, this is equivalent to showing that nearly all of the mass of $\mathcal{D}_f^\lambda$ lies in $\cup_{\mathbf{y} \in \mathbb{H}_d} B_d(\mathbf{y}, \frac{1}{4})$. We write

$$\begin{aligned} Z_\lambda^f &= \int_{B_d(0, R)} \exp(-\lambda f(\mathbf{x})) \, d\mathbf{x} \\ &\geq \sum_{\mathbf{y} \in \mathbb{H}_d} \int_{B_d(\mathbf{y}, \frac{1}{4})} \exp(-\lambda f(\mathbf{x})) \, d\mathbf{x} \\ &= \sum_{\mathbf{y} \in \mathbb{H}_d} \exp(-\lambda h(\mathbf{y})) \cdot \int_{B_d(\mathbf{y}, \frac{1}{4})} \exp(-s\lambda g(\mathbf{x})) \, d\mathbf{x} \end{aligned}$$

$$(1)$$

Let $\widehat{Z_\lambda^f}$ denote this last expression. We will argue that $Z_\lambda^f \leq (1 + \exp(-\Omega(d)))\widehat{Z_\lambda^f}$. We write $Z_\lambda^f$ as

$$\int_{B_d(0,R)} \exp(-\lambda f(\mathbf{x})) \, d\mathbf{x} \leq \sum_{\mathbf{y} \in \mathbb{H}_d} \int_{B_d(\mathbf{y}, \frac{1}{2})} \exp(-\lambda f(\mathbf{x})) \, d\mathbf{x} + \int_{B_d(0,R)} \exp(-\lambda M) \, d\mathbf{x}$$

$$\leq \underbrace{\sum_{\mathbf{y} \in \mathbb{H}_d} \exp(-\lambda h(\mathbf{y})) \cdot \int_{B_d(\mathbf{y}, \frac{1}{2})} \exp(-s\lambda g(\mathbf{x})) \, d\mathbf{x}}_{(A)} + \underbrace{\int_{B_d(0,R)} \exp(-\lambda M) \, d\mathbf{x}}_{(B)}$$

A simple calculation, formalized as Lemma 16 in supplementary material shows that the integral $\int_{B_d(\mathbf{y},\frac{1}{2})} \exp(-s\lambda g(\mathbf{x}))\,\mathrm{d}\mathbf{x}$ is within $(1 + 2\exp(-d))$ of $\int_{B_d(\mathbf{y},\frac{1}{4})} \exp(-s\lambda g(\mathbf{x}))\,\mathrm{d}\mathbf{x}$ for $s\lambda > 16d$.

This implies that the term $(A)$ above is at most $(1 + 2\exp(-d))\widehat{Z_\lambda^f}$. To bound the second term $(B)$ above, we argue that a ball of radius $\frac{1}{8}$ around any single vertex $\mathbf{y}$ of the hypercube contributes significantly more than than the term $(B)$. Indeed

$$\int_{B_d(\mathbf{y},\frac{1}{8})} \exp(-\lambda f(\mathbf{x}))\,\mathrm{d}\mathbf{x} \geq \int_{B_d(\mathbf{y},\frac{1}{8})} \exp(-\lambda(h(\mathbf{y}) + \frac{s}{8}))\,\mathrm{d}\mathbf{x}$$

$$\geq \exp(-\lambda(d + \frac{M}{4})) \cdot (\frac{1}{8})^d \int_{B_d(\mathbf{0},1)} \mathrm{d}\mathbf{x} \quad \geq \exp(-\lambda(\frac{3M}{4})) \cdot (\frac{1}{8})^d \int_{B_d(\mathbf{0},1)} \mathrm{d}\mathbf{x}$$

Whereas,

$$\int_{B_d(0,R)} \exp(-\lambda M)\,\mathrm{d}\mathbf{x} = \exp(-\lambda M) \cdot R^d \cdot \int_{B_d(\mathbf{0},1)} \mathrm{d}\mathbf{x}.$$

Thus $(B)/\widehat{Z_\lambda^f}$ is at most $\exp(-\lambda M/4 + d\ln 8R)$. Under the assumptions on $\lambda$, it follows that $(B)$ is at most $\exp(-\Omega(d))$ times $\widehat{Z_\lambda^h}$. In other words, we have shown that $Z_\lambda^f$ is at most $(1 + \exp(-\Omega(d)))\widehat{Z_\lambda^h}$.

□

*Remark* 4. The equivalence of sampling extends immediately to Wasserstein distance. Indeed given a sampler for $\mathcal{D}_h^{\lambda,\mathbb{H}_d}$, one gets a Wasserstein sampler for $\widehat{\mathcal{D}_f^\lambda}$ by sampling from a simple isotropic log-concave distribution. A Wasserstein sampler for a ball suffices for this. Since $\mathcal{W}(P,Q)$ is bounded by the diameter times the statistical distance, this gives a $\eta + \exp(-\Omega(d))$ Wasserstein sampler for $\mathcal{D}_f^\lambda$. Similarly, a $\eta$ Wasserstein sampler for $\mathcal{D}_f^\lambda$ conditioned on the support of $\widehat{\mathcal{D}_f^\lambda}$ immediately yields an $O(\eta)$-sampler for $\mathcal{D}_h^{\lambda,\mathbb{H}_d}$. Moreover, it is easy to check that this conditioning is on a constant probability event as long as $\eta < \frac{1}{16}$.

## 4.1 Optimization can be Easier than Sampling

Given the reduction from the previous section, there are many options for a starting discrete problem to apply the reduction. We will start from one of the most celebrated NP-hard problems. The NP-hardness of Hamiltonian Cycle dates back to Karp [1972].

**Theorem 5** (Hardness of HAMCYCLE). *Given a constant-degree graph $G = (V, E)$, it is NP-hard to distinguish the following two cases.*

YES **Case:** *$G$ has a Hamiltonian Cycle.*

NO **Case:** *$G$ has no Hamiltonian Cycle.*

We can then amplify the gap between the number of long cycles in the two cases.

**Theorem 6** (#CYCLE hardness). *Given a constant-degree graph $G = (V, E)$ and for $L = |V|/2$, it is NP-hard to distinguish the following two cases.*

YES **Case:** *$G$ has at least $1 + 2^L$ simple cycles of length $L$.*

NO **Case:** *$G$ has exactly one simple cycle $C^{(planted)}$ of length $L$, and no longer simple cycle.*

*Moreover, $C^{(planted)}$ can be efficiently found in polynomial time.*

The proof of the above uses a simple extension of a relatively standard reduction (see e.g. Vadhan [2002]) from Hamiltonian Cycle. Starting with an instance $G_1$ of Hamiltonian Cycle, we replace each edge by a two-connected path of length $t$, for some integer $t$. For $L = nt$, this gives us $2^L$ cycles of length $L$ for every Hamiltonian cycle in $G_1$. Moreover, any simple cycle of length $L$ must correspond to a Hamiltonian Cycle in $G_1$. We add to $G$ a simple cycle of length $L$ on a separate set of $L$ vertices. This "planted" cycle is easy to find, since it forms its own connected component of size $L$. A full proof is deferred to supplementary material.

Armed with these, we form a function on the hypercube in $d = |E'|$ dimensions such that optimizing it is easy, but sampling is hard.

**Theorem 7.** *There exists a function* $h : \mathbb{H}_d \to [0, d]$ *satisfying the following properties.*

**Efficiency:** $h$ *can be computed efficiently on* $\mathbb{H}_d$.

**Easy Optimization:** *One can efficiently find a particular minimizer* $\mathbf{y}^{(planted)}$ *of* $h$ *on* $\mathbb{H}_d$.

**Sampling is hard:** *Let* $\lambda \geq 2d$. *It is NP-hard to distinguish the following two cases, for* $L = \Omega(d)$:

> YES **Case:** $\Pr_{\mathbf{y} \sim \mathcal{D}_h^{\lambda, \mathbb{H}_d}}[\mathbf{y} = \mathbf{y}^{(planted)}] \leq \frac{1}{2^L}$
>
> NO **Case:** $\Pr_{\mathbf{y} \sim \mathcal{D}_h^{\lambda, \mathbb{H}_d}}[\mathbf{y} = \mathbf{y}^{(planted)}] \geq 1 - \frac{1}{2^L}$
>
> *In particular this implies that* $1 - \frac{1}{2^{L-2}}$-*sampling from* $\mathcal{D}_h^{\lambda, \mathbb{H}_d}$ *is NP-hard.*

*Proof.* Let $G = (V, E)$ a graph produced by Theorem 6 and let $d = |E|$. A vertex $\mathbf{y}$ of the hypercube $\mathbb{H}_d$ is easily identified with a set $E_{\mathbf{y}} \subset E$ consisting of the edges $\{e \in E : y_{e'} = 1\}$. The function $h_1(\mathbf{y})$ is equal to zero if $E_{\mathbf{y}}$ does not define a simple cycle and is equal to the length of the cycle otherwise. To convert this into a minimization problem, we define $h(\mathbf{y}) = d - h_1(\mathbf{y})$. It is immediate that a minimizer of $h$ corresponds to a longest simple cycle in $G$.

Given a vertex $\mathbf{y}$, testing whether $E_{\mathbf{y}}$ is a simple cycle can done efficiently, and the length of the cycle is simply $|E_{\mathbf{y}}|$. This implies that $h$ can be efficiently computed on $\mathbb{H}_d$. Further, since we can find the planted cycle in $G$ efficiently, we can efficiently construct a minimizer $\mathbf{y}^{(planted)}$ of $h$.

Suppose that $G$ has at least $(2^L + 1)$ cycles of length $L$. In this case, the distribution $\mathcal{D}_h^{\lambda, \mathbb{H}_d}$ restricted to the minimizers is uniform, and thus the probability mass on a specific minimizer $\mathbf{y}^{(planted)}$ is at most $\frac{1}{2^L + 1}$. This also therefore upper bounds the probability mass on $\mathbf{y}^{(planted)}$ in the $\mathcal{D}_h^{\lambda, \mathbb{H}_d}$.

On the other hand, suppose that planted cycle is the unique longest simple cycle in $G$. Thus the probability mass on $\mathbf{y}^{(planted)}$ is at least $\exp(\lambda(d - L))/Z_h^{\lambda, \mathbb{H}_d}$. Since every other cycle is of length at most $L - 1$, and there are at most $2^d$ cycles, it follows that $\frac{Z_h^{\lambda, \mathbb{H}_d}}{\exp(\lambda(d-L))} \leq 1 + \frac{2^d}{\exp(\lambda)}$. For $\lambda \geq 2d$, this ratio is $(1 + \exp(-d)) \leq (1 - \frac{1}{2^d})^{-1}$. The claim follows. $\square$

We can now apply Theorem 4 to derive the following result.

**Theorem 8.** *There is a family* $\mathcal{F}$ *of functions* $f : B_d(0, R) \to \mathbb{R}$ *such that the following hold.*

**Efficiency:** *Every* $f \in \mathcal{F}$ *is computable in time* $poly(d)$.

**Easy Optimization:** *An optimizer of* $f$ *can be found in time* $poly(d)$

**Sampling is NP-hard:** *For* $\lambda \geq 2d$ *and* $\eta < 1 - \exp(-\Omega(d))$, *there is no efficient* $\eta$-*sampler for* $\mathcal{D}_f^{\lambda}$ *unless* $NP = RP$.

*Remark* 5. In the statement above, efficiently computable means that given a $t$-bit representation of $\mathbf{x}$, one can compute $f(\mathbf{x})$ to $t$ bits of accuracy in time $poly(d, t)$.

*Remark* 6. The easy optimization result above can be considerably strengthened. We can ensure that $\mathbf{0}$ is the optimizer of $f$ and that except with negligible probability, gradient descent starting at a random point will converge to this minimizer. Further, one can ensure that all local minima are global and that $f$ is strict-saddle. Thus not only is the function easy to optimize given the representation, black box oracle access to $f$ and its gradients suffices to optimize $f$. We defer details to the supplementary material.

*Remark* 7. The hardness of sampling holds also for Wasserstein distance $\frac{1}{16}$, given Remark 4.

## 5 A Sharp Threshold for $\lambda$

We start with the following threshold result for sampling from the Gibb's distribution on independent sets due to Weitz [2006], Sly and Sun [2012].

**Theorem 9.** *For any* $\Delta \geq 6$, *there is a threshold* $\lambda_c(\Delta) > 0$ *such that the following are true.*

**FPRAS for small** $\lambda$: *For any $\lambda < \lambda_c(\Delta)$, the problem of sampling independent sets with $\Pr[I] \propto \exp(\lambda|I|)$ on $\Delta$-regular graphs has a fully polynomial time approximation scheme.*

**NP-hard for large** $\lambda$: *For any $\lambda > \lambda_c(\Delta)$, unless $NP = RP$, there is no fully polynomial time randomized approximation scheme for the problem of sampling independent sets with $\Pr[I] \propto \exp(\lambda|I|)$ on $\Delta$-regular graphs.*

In the supplementarly material, we show that this implies the following result.

**Theorem 10.** *There is a family $\mathcal{F}$ of efficiently computable functions $f : B_d(0, R) \to \mathbb{R}$ such that the sampling problem has a sharp computational threshold. There is a constant $\lambda_c > 0$ such that for any $\frac{1}{d} < \lambda < \lambda_c$, there is a poly($d/\eta$)-time $\eta$-sampler for the distribution $\mathcal{D}_f^\lambda$. On the other hand, for any $\lambda > \lambda_c$, there is a constant $\eta' > 0$ such that no polynomial time algorithm $\eta'$-samples from $\mathcal{D}_f^\lambda$ unless $NP = RP$.*

# 6 Related Work

The problems of counting solutions and sampling solutions are intimately related, and well-studied in the discrete case. The class #P was defined in Valiant [1979], where he showed that the problem of computing the permanenet of matrix was complete for this class. This class has been well-studied, and Toda [1991] showed efficient exact algorithms for any #P-complete problem would imply a collapse of the polynomial hierarchy. Many common problems in #P however admit efficient approximation schemes, that for any $\varepsilon > 0$, allow for a randomized $(1 + \varepsilon)$-approximation in time polynomial in $n/\varepsilon$. Such *Fully Polynomial Randomized Approximation Schemes* (FPRASes) are known for many problems in #P, perhaps the most celebrated of them being that for the permananet of a non-negative matrix [Jerrum et al., 2004].

These FPRASes are nearly always based on Markov Chain methods, and their Metropolis-Hastings [Metropolis et al., 1953, Hastings, 1970] variants. These techniques have been used both in the discrete case (e.g. Jerrum et al. [2004]) and the continuous case (e.g. Lovasz and Vempala [2006]). The closely related technique of Langevin dynamics [Rossky et al., 1978, O. Roberts and Stramer, 2002, Durmus and Moulines, 2017] and its Metropolis-adjusted variant are often faster in practice and have only recently been analyzed.

## Footnotes

[1]As defined, this function is Lipschitz but not smooth. It can be easily modified to a 2-Lipschitz, $2/\varepsilon$-smooth function by replacing its linear behavior in the ball by an appropriately Huberized version.

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
