[Supplementary Material]

## A  Efficient Net Construction

We start with the existence of error correcting codes for worst case error (e.g. Justesen [1972]).

**Theorem 11.** *There exists a universal constant $\alpha > 0$, and an efficiently computable mapping $\psi : \{0,1\}^d \to \{0,1\}^{4d}$ such that for any $\mathbf{y} \neq \mathbf{y}' \in \{0,1\}^d$, the Hamming distance $d_H(\psi(\mathbf{y}), \psi(\mathbf{y}')) \geq 4\alpha d$. Further, there is an efficiently computable mapping $\mathtt{Decode} : \{0,1\}^{4d} \to \{0,1\}^d$ such that for any $\mathbf{y} \in \{0,1\}^D$ and any $\mathbf{w}$ satisfying $d_H(\psi(\mathbf{y}), \mathbf{w}) \leq \alpha d$, $\mathtt{Decode}(\mathbf{w})$ returns $\mathbf{y}$.*

Armed with this, we prove the desired net construction.

**Theorem 12.** *There is an absolute constant $\varepsilon > 0$ and an efficiently computable map $\phi : \{0,1\}^d \to B_d(\mathbf{0}, 1)$ such that (a) $\|\phi(\mathbf{y}) - \phi(\mathbf{y}')\| \geq 4\varepsilon$ and (b) $\mathbf{x} \in B_d(\phi(\mathbf{y}), 2\varepsilon)$, we can efficiently recover $\mathbf{y}$.*

*Proof.* The map $\phi(\mathbf{y})$ will be constructed by embedding $\psi(\mathbf{y})$ into $B_d(\mathbf{0}, 1)$. We first interpret $\psi(\mathbf{y})$ as a vector $\hat{\mathbf{y}}$ in $\{0, 1, \ldots, 15\}^d$, with the $i$th co-ordinate of $\hat{\mathbf{y}}$ being defined by the 4 bits $\psi(\mathbf{y})_{4i-3}, \psi(\mathbf{y})_{4i-2}, \psi(\mathbf{y})_{4i-1}, \psi(\mathbf{y})_{4i}$. The $i$th co-ordinate of $\phi(\mathbf{y})$ is set to $\hat{\mathbf{y}}_i / 15\sqrt{d}$. It is easy to check that $\phi(\mathbf{y})$ as defined lies in the unit ball. Consider $\mathbf{y}, \mathbf{y}' \in \{0,1\}^d$ such that $d_H(\psi(\mathbf{y}), \psi(\mathbf{y}')) \geq 4\alpha d$. This means that $\hat{\mathbf{y}}$ and $\hat{\mathbf{y}'}$ differ in at least $\alpha d$ locations. This in turn means that $\|\phi(\mathbf{y}) - \phi(\mathbf{y}')\| \geq \sqrt{\alpha}/15$.

Let $\mathbf{x} \in B_d(\phi(\mathbf{y}), \varepsilon)$. By Markov's inequality, $|\{i : |\phi(\mathbf{y})_i - \mathbf{x}_i| \geq \frac{1}{30\sqrt{d}}\}|$ is at most $900\varepsilon^2 d$. Thus if we greedily decode each co-ordinate of $\mathbf{x}$ and expand that to a $4d$-bit vector, the decoding will be correct except in $3600\varepsilon^2 d$ locations. Letting $\varepsilon = \sqrt{\alpha}/120$, and using properties of $\psi$, the claim follows. $\qquad\square$

## B  Sampling to Optimization

The following folklore theorem shows that sampling for high $\lambda$ implies approximate optimization.

**Lemma 13.** *[Sampling Implies Optimization: Generic] Let $f : B_d(0, R) \to \mathbb{R}$ and suppose that for some $a \in \mathbb{R}$, the level set $L_a = \{\mathbf{x} \in B_d(0, R) : f(\mathbf{x}) \geq a\}$ is measurable. Then for any $\varepsilon > 0$, and for $\lambda \geq \frac{\ln \frac{1}{\delta} + \ln \frac{\mathrm{vol}_d(B_d(0,R))}{\mathrm{vol}_d(L_a)}}{\varepsilon}$,*

$$\Pr_{\mathbf{x} \sim \mathcal{D}_\lambda^f}[f(\mathbf{x}) \geq a + \varepsilon] \leq \delta.$$

*Proof.* We write

$$\Pr_{\mathbf{x} \sim \mathcal{D}_\lambda^f}[f(\mathbf{x}) \geq a + \varepsilon] \leq \frac{\exp(-\lambda(a + \varepsilon)) \cdot \mathrm{vol}_d(B_d(0, R))}{Z_f^\lambda}$$

$$\leq \frac{\exp(-\lambda(a + \varepsilon)) \cdot \mathrm{vol}_d(B_d(0, R))}{\exp(\lambda a) \cdot \mathrm{vol}_d(L_a)}$$

$$= \frac{\exp(-\lambda \varepsilon)}{\frac{\mathrm{vol}_d(L_a)}{\mathrm{vol}_d(B_d(0,R))}}$$

$$\leq \delta.$$

$\qquad\square$

**Theorem 14.** *[Sampling Implies Optimization: Lipschitz $f$] Let $f : B_d(0, R) \to \mathbb{R}$ be $L$-Lipschitz and let $f^\star = \min_{\mathbf{x} \in B_d(0,R)} f(\mathbf{x})$. Then for any $\varepsilon > 0$ and for $\lambda \geq \frac{2d \ln \frac{4LR}{\varepsilon} + 2 \ln \frac{1}{\delta}}{\varepsilon}$,*

$$\Pr_{\mathbf{x} \sim \mathcal{D}_{)f^\lambda}}[f(\mathbf{x}) \geq f^\star + \varepsilon] \leq \delta.$$

*Proof.* Set $a = \varepsilon/2$, and note that by Lipschitz-ness of $f$, the level set $L_a$ contains a ball of radius $\varepsilon/2L$ around the optimizer, so that $\frac{\mathrm{vol}_d(L_a)}{\mathrm{vol}_d(B_d(0,R))}$ is at least $(\varepsilon/4LR)^d$. Applying Theorem 13, the claim follows. $\qquad\square$

**Theorem 15** (Sampling Implies Optimization: Smooth $f$). *Let $f : B_d(0, R) \to \mathbb{R}$ be $\beta$-smooth and let $f^\star = \min_{\mathbf{x} \in B_d(0,R)} f(\mathbf{x})$ be attained in the interior of $B_d(0, R)$. Then for any $\varepsilon > 0$ and for*
$$\lambda \geq \frac{d \ln \frac{\beta R^2}{\varepsilon} + 2 \ln \frac{1}{\delta}}{\varepsilon},$$

$$\Pr_{\mathbf{x} \sim \mathcal{D}_) f^\lambda}[f(\mathbf{x}) \geq f^\star + \varepsilon] \leq \delta.$$

*Proof.* Let the mimimum $f^\star$ be attained at $\mathbf{x}^\star$; by assumption $\nabla f$ at $\mathbf{x}^\star$ is zero, and by $\beta$-smoothness, the level set $L_{\varepsilon/2}$ contains a ball of radius $(2\varepsilon/\beta)^{\frac{1}{2}}$ around $\mathbf{x}^\star$. The claim follows. $\square$

We note that $\delta = \frac{1}{2}$ suffices to ensure that efficient samplability implies efficient approximate optimization, since we can run the sampler multiple times.

## C  Deferred Proofs

### C.1  Incomplete Gamma Functions

The following was used in the proof of Theorem 4.

**Lemma 16.** *Let $d$ be a positive integer and $\alpha \geq 16d$. Then*

$$\int_0^{\frac{1}{2}} \exp(-\alpha r) r^{d-1} \, \mathrm{d}r \leq (1 + 2\exp(-d)) \int_0^{\frac{1}{4}} \exp(-\alpha r) r^{d-1} \, \mathrm{d}r$$

*Proof.* By definition of the Incomplete Gamma function, for any $a \geq 0$,

$$\int_0^a \exp(-\alpha r) r^{d-1} \, \mathrm{d}r = \frac{1}{\alpha^d} \cdot \int_0^{a\alpha} \exp(-s) s^{d-1} \, \mathrm{d}s$$
$$= \frac{\gamma(d, a\alpha)}{\alpha^d}.$$

Translated to this language, we wish to bound

$$\frac{\gamma(d, \alpha/2)}{\gamma(d, \alpha/4)}.$$

We now bound

$$1 - \frac{\gamma(d, \alpha/4)}{\gamma(d, \alpha/2)} \leq 1 - \frac{\gamma(d, \alpha/4)}{\Gamma(d)}$$
$$= \Pr_{X \sim \Gamma(d,1)}[X \geq \alpha/4]$$
$$\leq \Pr_{X \sim \Gamma(d,1)}[X \geq d + 3d]$$
$$\leq \exp(-d).$$

Here in the first step, we have used the fact that the Incomplete Gamma function is the cdf of the corresponding Gamma distribution. The last step uses standard tail inequalities for sub-gamma distributions from Boucheron et al. [2013, Section 2.4].

We have thus shown that $\frac{\gamma(d,\alpha/4)}{\gamma(d,\alpha/2)} \geq 1 - \exp(-d)$. Thus $\gamma(d, \alpha/2) \leq (1 - \exp(-d))^{-1} \gamma(d, \alpha/4)$. For $d \geq 1$, $(1 - \exp(d))^{-1} \leq (1 + 2\exp(-d))$ and the claim follows. $\square$

## D  Count Gap Amplification for Cycles

Theorem 6 follows from the NP-hardness of Hamiltonian cycle and the following reduction.

**Theorem 17.** *Given a graph $G = (V, E)$ and an integer $k$, there is a polynomial time algorithm that outputs a graph $G' = (V', E')$ such that*

Figure 2: (Left) An example of a function $h$ for $d = 2$. (Right) The corresponding function $f$ that results from the transformation, for $M = 4, R = 2$. Note that we create a new minimizer at $\mathbf{0}$.

**"Completeness":** *If $G$ has a Hamiltonian cycle, then $G'$ has at least $1 + 2^{tn}$ simple cycles of length $nt$.*

**"Soundness":** *If $G$ has no Hamiltonian cycle, then $G'$ has exactly one cycle of length $nt$, and no longer simple cycles.*

**Efficiency:** *It is easy to find one cycle of length $nt$ in $G'$.*

*Proof.* The reduction replaces each edge of $G$ by a path of length $t$ with each edge on the path being duplicated. In addition, we add a new set of $tn$ vertices that form a cycle, to ensure that we always have once cycle of length $tn$. We give more details next.

Let $G = (V, E)$ and let $e = (u, v) \in E$. Our new vertex set $V' = V_1 \cup V_2$, where $V_1 = V \cup \{e_i : e \in E, i \in [t-1]\}$ and $V_2 = \{w_1, \ldots, w_{nt}\}$. The vertices in $V_2$ form a simple cycle, i.e. $E_2 = \{(w_i, w_{i+1} : i \in [nt-1])\} \cup \{(w_{nt}, w_1)\}$. For every edge $e = (u, v) \in E$, $E'$ contains two copies of each edge set $E_1^e = \{(u, e_1), (e_t, v)\} \cup \{(e_i, e_{i+1}) : i \in [t-1]\}$.

It is easy to see that $G'$ always has one cycle of length $nt$ consisting of $E_2$ that can be found efficiently. Whenever $G$ has a Hamiltonian cycle, we can form a cycle of length $nt$, by following the paths corresponding to the edges used in the Hamilitonian cycle in $G$. At each step, we have a choice of two edges to choose from, since $E'$ has parallel edges. This gives us $2^{nt}$ such cycles, proving the "completeness" part of the theorem.

Finally note that for any simple cycle of length $nt$ on $V_1$, the projection of the cycle onto the vertices in $V$ is a simple cycle of length $n$, i.e. a Hamiltonian cycle. This completes the proof of the "soundness" part of the theorem. □

# E  Stronger Optimizability

In this section, we show that the separation between optimization and sampling holds even for stronger notions of $f$ being optimizable.

**Theorem 18.** *There is a family $\mathcal{F}$ of functions $f : \mathbb{R}^d \to \mathbb{R}$ such that the following hold.*

**Efficiency:** *Each $f \in \mathcal{F}$ is computable in time $poly(d)$.*

**Easy Optimization:** *The zero vector $\mathbf{0}$ is a global optimizer of $f$. Further, $f$ satisfies strict saddle, and a randomly initialized gradient descent algorithm will converge to $\mathbf{0}$ with high probability.*

**Hard to Sample:** *For $\lambda \geq 2d$, there is no $1 - \exp(-\Omega(d))$-sampler for $\mathcal{D}_f^\lambda$ unless $NP = RP$.*

*Proof.* The proof is very similar to that of Theorem 8. We will highlight the relevant changes. First, we redefine $h$ slightly: $h$ now takes the value 0 whenever $E_{\mathbf{y}}$ defines a simple cycle of length $L$, and takes the value $d$ otherwise. Note that it is NP-hard to determines if $h^{-1}(0)$ has size 1 or $2^L + 1$.

The construction from $h$ to $f$ is similar to that in Theorem 4 with some variation. The function $f_1$ takes the value $M$, except in small balls around $h^{-1}(0)$ and a new minima at 0. In addition, we add a linear term. Recalling the definition of $\mathrm{round}(\mathbf{x})$ and $g(\mathbf{x})$ from the proof of Theorem 4, we define $f(\mathbf{x})$ :

$$
f(\mathbf{x}) = \begin{cases}
\|\mathbf{x}\| - \sqrt{d} + 16(M + \sqrt{d}) \cdot g(\mathbf{x}) & \text{if } h(\mathrm{round}(\mathbf{x})) = 0 \text{ and } g(\mathbf{x}) \leq \frac{1}{16} \\
\|\mathbf{x}\| + 16M \cdot \|\mathbf{x}\| & \text{if } \|\mathbf{x}\| \leq \frac{1}{16} \\
\|\mathbf{x}\| + M & \text{otherwise}
\end{cases}
$$

It is easy to verify that $f$ satisfies the following properties:

- $f(\mathbf{0}) = 0$. For $\mathbf{y} \in \mathbb{H}_d$, $h(\mathbf{y}) = 0 \Leftrightarrow f(\mathbf{y}) = 0$.

- $f$ is efficiently computable.

- $f$ is continuous and $O(M)$-Lipschitz.

- Outside of $\cup_{\mathbf{y}:h(\mathbf{y}=0)\wedge\mathbf{y}=\mathbf{0}} B_d(\mathbf{y}, \frac{1}{16})$, $f$ is equal to $M + \|\mathbf{x}\|$.

It follows that the gradient at most points points towards the origin. Thus if the line joining the initial point of a gradient descent and $\mathbf{0}$ avoids hitting $\cup_{\mathbf{y}:h(\mathbf{y}=0)} B_d(\mathbf{y}, \frac{1}{16})$, gradient descent will converge to the origin. Since this happens with high probability, it follows that GD will succeed on $f$ w.h.p. While $f$ as defined is not twice differentiable, convolving $f$ with a small Gaussian gives us a function that is infinitely differentiable, and whose Lipschitz constant, and behavior with respect to gradient descent does not significantly change.

The hardness of sampling proof is essentially unchanged, and thus omitted. $\square$

# F   Sharp Threshold Proof

We prove the following result.

**Theorem 19.** *There is a family $\mathcal{F}$ of functions $f : B_d(0, R) \to \mathbb{R}$ such that the following hold.*

**Efficiency:** *Every $f \in \mathcal{F}$ is computable in time $poly(d)$.*

**Sampling has a threshold:** *There is a constant $\lambda_c > 0$ such that for any $\frac{1}{d} < \lambda < \lambda_c$, there is a $poly(d/\eta)$-time $\eta$-sampler from from the distribution $\mathcal{D}_f^\lambda$. On the other hand, for $\lambda > \lambda_c$, there is a constant $\eta' > 0$ such that no polynomial time algorithm $\eta'$-samples from $\mathcal{D}_f^\lambda$ unless $NP = RP$.*

*Proof.* For a graph $G = (V, E)$, the function $h$ on the hypercube $\mathbb{H}_d$ is defined in the natural way for $d = |V|$. We identify $\mathbf{y}$ with $V_{\mathbf{y}} = \{v_i : \mathbf{y}_i = 1\}$, and set $h(\mathbf{y}) = d - |V_{\mathbf{y}}|$ if $V_{\mathbf{y}}$ is an independent set in $G$ and to $d$ otherwise. We then apply Theorem 4, with $R = 2\sqrt{d}$ and $M = 4d^2 \ln 24R$. For this value of $M$, the approximate equivalence between sampling from $\exp(\lambda|I|)$ and sampling from $\exp(-\lambda f)$ holds for $\lambda \geq \frac{1}{d}$. The claim follows. $\square$