[Reviews · NeurIPS 2019]

Reviewer 1



For example, the only proof which is more than a few lines long is for the result which shows that there exist some problem where sampling is hard and optimization is easy. Here is an alternative proof: take any problem where sampling is hard, change the value at 0 to be f(0) = -M. optimization became easy, but sampling is still equally hard.

Reviewer 2



his paper considers two natural problems that arise in machine learning (i) maximizing a function f(x) and (ii) sampling approximately from the distribution e^(-f(x)). The goal is to show that under some situations, one of these problems is easy and the other is hard. To show that optimization can be harder than sampling, the construction hides the solution of an NP-hard problem as a small bump in a mostly flat function. Thus, approximate sampling is easy (the distribution is mostly uniform), but optimization would result in solving an NP-hard problem. To show that sampling can be harder than optimization, the construction amplifies the number of solutions of an NP-hard problem and plants an additional simple solution, and then encodes this into a function that is flat in many places, but has bumps at every possible solution of the NP-hard problem. Optimization is as easy as finding the planted simple solution, but, intuitively, sampling requires finding many of the hard solutions. In general, the proofs are elegant and intuitive, and the writing is very clear. However, there is an issue with the particular definition of approximate sampling (being close in total variation distance) used throughout the paper. Because these separations are computational, we have to be wary of defining problems that can only be solved by computing real numbers. In the case of total variation distance, the distance between a discrete and a continuous distribution is always 1, meaning that no computer can approximately sample (with respect to total variation distance) from a continuous distribution with a finite number of bits. One possible resolution to this problem is to use a different notion of approximation. Wasserstein distance, for example, allows for non-trivial distances between discrete and continuous distributions, and has been employed in reductions between optimization and sampling problems [TD19]. Based on the similarity of that paper to this one, it seems like it should not be too hard to show the exact same results in this paper but under a Wasserstein distance-type approximation. Thus, I am inclined to accept this paper, with the recommendation to the authors that they change the definition of approximate sampling to something that allows for non-trivial distances between discrete and continuous distributions. Typos: - Line 163: min(0, |y - x| - 2\epsilon) should be min(0, |\phi(y) - x| - 2\epsilon). References: [TD19] C. Tosh and S. Dasgupta. The relative complexity of maximum likelihood estimation, MAP estimation, and sampling. COLT 2019.

Reviewer 3



Summary --------------- This paper explores the computational complexity relation between sampling and optimization. More precisely given a function f(x) defined over a set X \subset of R^d, we can define the following problems: - Optimization: find x' such that f(x') <= f(x*) + epsilon where x* is the minimizer of x'. - Sampling: approximately sample y from X from the probability distribution with density p(y) \prop exp(-f(y)). When f(x) is a convex function of x then it is known that optimization is computationally equivalent with sampling and both can be solved in polynomial time. When f(x) is non-convex though too little is known for the relation of these two problems. A recent work by Ma et al shows an example where sampling is easy but optimization is hard in the oracle model of Nesterov. This paper enriches our understanding of sampling vs optimization in two ways: (1) it strengths the example provided by Ma et al. by showing that there are cases where the sampling is easy but optimization is NP-hard, (2) it shows that the opposite is true too, optimization can be easy but sampling NP-hard. Observe that NP-hardness is a very strong notion of hardness from computational complexity theory and in particular it implies hardness in the oracle model. Strengths -------------------- - The understanding of the relation between sampling and optimization in the non-convex case is a very fundamental and relevant question. - The results are the strongest possible in this direction and they rule out for example the possible use of second or higher order methods. - The proofs provide a clean but non-trivial use of complexity of combinatorial problems in a continuous space. I believe that the techniques provided in this paper can be of independent interest for showing more NP-hardness results for continuous non-convex problems. Weaknesses ------------------- Strictly technically speaking the proofs use heavy machinery from known results in complexity theory but are non very technical themselves. Summary of Recommendation ----------------------------- The problem explored in this paper is fundamental and relevant, the results are strong and the presentation very good so I strongly recommend acceptance.

[Author Response · NeurIPS 2019]

We thank the reviewers for their insightful comments. We will address all these comments when updating the paper. Below we will address some of the specific concerns raised by the reviewers.

Reviewer 2 correctly pointed out that the results could be made more robust by considering the Wasserstein distance between distributions instead of statistical distance. Given that our hardness results are in fact for the discretized versions of the sampling problem, we can indeed extend all our results to the Wasserstein distance. We have verified the details and will update the paper to address this.

Reviewer 1 had concerns about the interesting-ness of the question. We will explain why we find the questions interesting. In the convex setting, the sampling and optimization problems have a very close connection. Many parts of modern ML deal with loss functions (or likelihood functions) that are not convex and in practice, we often want to solve sampling and optimization problems. For completely arbitrary $f$, these problems are likely unrelated. One might hope that for "natural" $f$, these two problems are not too different from a computational point of view, and one might hope to explain this by exploiting various properties that natural $f$ satisfy. For example $f$'s of interest may be efficiently computable, continuous, and smooth. Separations of the kind proved in our work show that just these conditions are not sufficient to establish any kind of computational equivalence. If for a specific problem, I want to argue equivalence, I would need to argue that the $f$'s I care about have some additional properties that rule out the kind of separation results established in our work. Additionally, the framework may be useful to further understand what additional properties of natural functions are needed to avoid such separations.

Reviewer 1 remarked that *"the only proof which is more than a few lines long is for the result which shows that there exist some problem where sampling is hard and optimization is easy."*

We believe that the simplicity of the arguments is a feature. Indeed the previous work by Ma et al. had a long and complicated proof of a weaker version of the first result in our work.

Reviewer 1 also gave an alternate argument for one of the results in the work : *"Here is an alternative proof: take any problem where sampling is hard, change the value at $0$ to be $f(0) = -M$. optimization became easy, but sampling is still equally hard."*

To our knowledge, we lacked the tools to prove that sampling is hard for some $f$'s, so that it is not obvious how to start with a problem where sampling is hard. One of our contributions is to provide the tools to do that. Further, the function $f$ defined by the reviewer is neither continuous nor Lipschitz and thus does not rule out equivalence of sampling and optimization for continuous and Lipschitz functions. More generally, our goal in this work was to establish a connection between the rich and well-developed area of complexity of discrete counting problems, and the question of sampling for continuous functions.

As suggested by Reviewer 3, we will add a section on future research directions.

[Meta-Review · NeurIPS 2019]

The paper demonstrates computational separation between sampling and optimization, specifically problems where sampling is easy but computing a global optima is hard, and vice versa. One nice aspect here is that the paper brings complexity-theoretic tools to the optimization community, which has been thinking about lower bounds mostly via information theoretic tools (Nesterov's black box model). One confusion that was clarified in the discussion is that the reductions used need to preserve continuity/Lipschitzness, which I recommend the authors emphasize in the paper. I also recommend that the results be translated to Wasserstein distance (as opposed to TV) as discussed in the reviews and the author response.